# Nuclear lipid droplets derive from a lipoprotein precursor and regulate phosphatidylcholine synthesis

Kamil Sołtysik[1], Yuki Ohsaki[1], Tsuyako Tatematsu[1], Jinglei Cheng[1] & Toyoshi Fujimoto [1]

The origin and physiological significance of lipid droplets (LDs) in the nucleus is not clear. Here we show that nuclear LDs in hepatocytes are derived from apolipoprotein B (ApoB)-free lumenal LDs, a precursor to very low-density lipoproprotein (VLDL) generated in the ER lumen by microsomal triglyceride transfer protein. ApoB-free lumenal LDs accumulate under ER stress, grow within the lumen of the type I nucleoplasmic reticulum, and turn into nucleoplasmic LDs by disintegration of the surrounding inner nuclear membrane. Oleic acid with or without tunicamycin significantly increases the formation of nucleoplasmic LDs, to which CTP:phosphocholine cytidylyltransferase α (CCTα) is recruited, resulting in activation of phosphatidylcholine (PC) synthesis. Perilipin-3 competes with CCTα in binding to nucleoplasmic LDs, and thus, knockdown and overexpression of perilipin-3 increases and decreases PC synthesis, respectively. The results indicate that nucleoplasmic LDs in hepatocytes constitute a feedback mechanism to regulate PC synthesis in accordance with ER stress.

---

[1] Department of Molecular Cell Biology and Anatomy, Nagoya University Graduate School of Medicine, Nagoya 466-8550, Japan. These authors contributed equally: Kamil Sołtysik, Yuki Ohsaki. Correspondence and requests for materials should be addressed to Y.O. (email: yohsaki@med.nagoya-u.ac.jp) or to T.F. (email: tfujimot@med.nagoya-u.ac.jp)

ipid droplets (LDs) exist widely in eukaryotic cells and are related to diverse cellular functions[1–3]. LDs are largely confined to the cytoplasm, but in some cell types, especially in hepatocytes, a relatively large number of LDs exist inside the nucleus[4,5]. We reported that nuclear LDs in hepatocytes are associated with the promyelocytic leukemia (PML) nuclear body and the intranuclear extension of the inner nuclear membrane (INM), or the type I nucleoplasmic reticulum (NR)[6]. The association with these bona fide nuclear structures suggested that nuclear LDs form by a mechanism different from cytoplasmic LDs, but it was not clear why nuclear LDs are abundant only in limited cell types and what function they have.

The abundance of nuclear LDs in hepatocytes led us to hypothesize that they may be related to the synthesis of very low-density lipoprotein (VLDL). In VLDL synthesis, two kinds of lumenal LDs are generated in the endoplasmic reticulum (ER) by the activity of microsome triglyceride transfer protein (MTP). They are primordial apolipoprotein B100 (ApoB)-containing particle and ApoB-free lumenal LDs, which give rise to mature VLDL in post-ER compartments[7–9]. MTP inhibition suppresses generation of these lumenal LDs and increases cytoplasmic LDs, which provide most lipids for VLDL synthesis[10].

We find that ApoB-free lumenal LDs accumulate under ER stress and generate large LDs in the type I NR lumen, which then relocate to the nucleoplasm through defects in the NR membrane. That is, LDs in the nucleoplasm are derived from a VLDL precursor in the ER lumen. Nucleoplasmic LDs that form by this surprising mechanism recruit CTP:phosphocholine cytidylyltransferase α (CCTα), the rate-limiting enzyme of the Kennedy pathway for phosphatidylcholine (PC) synthesis[11], and increase de novo PC synthesis. Perilipin-3 competes with CCTα in binding to nucleoplasmic LDs. Thus, knockdown of perilipin-3 upregulates PC synthesis by increasing nucleoplasmic LD-bound CCTα, whereas overexpression of perilipin-3 decreases CCTα in nucleoplasmic LDs and suppresses PC synthesis. The result indicates that nucleoplasmic LDs in hepatocytes constitute a feedback mechanism to regulate PC synthesis in accordance with the level of ER stress.

Hereafter in this manuscript, for clear distinction of LDs, LDs in the nucleoplasm and LDs in the type I NR lumen will be referred to as nucleoplasmic LDs and NR-lumenal LDs, respectively. LDs in the nuclear region will be referred to generally as nuclear LDs, when nucleoplasmic LDs and NR-lumenal LDs are not treated separately.

## Results

**MTP activity is essential for nuclear LD formation.** Incubation with 0.4 mM oleic acid (OA) increases both nuclear and cytoplasmic LDs in hepatocarcinoma cell lines[6]. We found that MTP inhibitors (MTPi), BAY 13-9952[12], and CP-346086[13], suppressed the OA-induced increase of nuclear LDs, but not that of cytoplasmic LDs in Huh7 (Fig. 1a). MTPi also reduced nuclear LDs in other hepatocarcinoma cell lines, HepG2 and McA-RH7777, but not in U2OS, which harbors nuclear LDs despite its osteosarcoma origin[6] (Supplementary Fig. 1a).

Knockdown of MTP by RNAi also suppressed the OA-induced increase of nuclear LDs, but not that of cytoplasmic LDs (Fig. 1b, Supplementary Fig. 1b). Co-transfection of MTP complementary DNA (cDNA) canceled the effect of RNAi, excluding the possibility of an off-target effect (Fig. 1b). Moreover, the MTP overexpression increased nuclear LDs significantly above the control level (Fig. 1b), suggesting that a disproportionate increase of MTP, which enhances the production of ApoB-free lumenal LDs[7,14], may enhance nuclear LD formation.

A similar imbalance between MTP and ApoB is known to occur when ApoB decreases in hepatocytes under ER stress[15,16]. We could reproduce the result by treating Huh7 with 0.4 mM OA and 5 μg/ml tunicamycin (OA/TM) (Supplementary Fig. 1c), and nuclear LDs increased in this condition (Fig. 1c). A higher concentration of OA (1.2 mM OA), a stronger ER stress than 0.4 mM OA[17], also increased nuclear LDs, but TM alone (TM) did not. The increase of nuclear LDs caused by OA/TM was suppressed by MTPi (Fig. 1c). OA/TM did not increase nuclear LDs in non-hepatic cell lines (Supplementary Fig. 1d).

The results indicated that the nuclear LD formation in hepatocytes may be correlated with ApoB-free lumenal LDs generated by MTP activity.

**LDs are present in the type I NR lumen.** Electron microscopy (EM) revealed that Huh7 treated with OA/TM harbors lumenal LDs not only in the ER, but also in the nuclear envelope and the type I NR (Fig. 2a). Many nucleoplasmic LDs were also observed (Supplementary Fig. 2a). NR-lumenal LDs were delineated clearly with 3,3'-diaminobenzidine (DAB) precipitates in cells expressing HRP-KDEL (Fig. 2b). NR-lumenal LDs were also present in cells treated with OA alone, albeit less frequently than in OA/TM-treated cells (Supplementary Fig. 2b).

Lumenal LDs also occurred in mouse hepatocytes in vivo after high-fat diet feeding and TM administration (Fig. 2c) and in mouse primary cultured hepatocytes treated with OA/TM (Supplementary Fig. 2c). The total nuclear LDs also increased in mouse hepatocytes in vivo (Fig. 2c).

Consistent with the EM result, the microsome of the OA/TM-treated Huh7 contained more triglycerides and cholesterol esters than that of untreated cell (Fig. 2d). In contrast, much less ApoB was in the OA/TM-treated microsome than in the control, indicating that NR-lumenal LDs in the OA/TM-treated cell are largely ApoB-free lumenal LDs (Fig. 2d). This conclusion was supported by the lack of ApoB labeling in the nucleus of OA- or OA/TM-treated Huh7 despite the presence of abundant nuclear LDs (Supplementary Fig. 2d).

The above result indicated that nuclear LDs observed by fluorescence microscopy contain nucleoplasmic LDs, NR-lumenal LDs, and cytoplasmic LDs within the type II NR (i.e., nuclear invagination) (Fig. 2e). By using lamin B1 receptor (LBR), a transmembrane protein of the INM, as a marker, LDs outside of LBR rings can be judged as nucleoplasmic LDs (Fig. 2f, Supplementary Fig. 2e). In contrast, LDs within LBR rings may be either NR-lumenal LDs or cytoplasmic LDs within the type II NR, but EM revealed the latter to be relatively scarce (Fig. 2g). The infrequency of the type II NR was also indicated by the scarcity of nuclear pore labeling within nuclei (Supplementary Fig. 2f). We therefore concluded that the majority of LDs observed within LBR rings are NR-lumenal LDs. By using this criterion, we confirmed that both NR-lumenal LDs and nucleoplasmic LDs increase in cells treated with OA or OA/TM, and that the increase in both LDs is suppressed by MTPi (Fig. 2f, Supplementary Fig. 2e). Soluble lumenal proteins, EGFP-KDEL and MTP-EGFP, gave results comparable to LBR (Supplementary Fig. 2g).

**NR-lumenal LDs are converted to nucleoplasmic LDs.** The above result suggested that NR-lumenal LDs may become nucleoplasmic LDs by a hitherto unknown mechanism. Consistent with this idea, ApoE and ApoCIII, apolipoproteins which bind to lumenal LDs and are normally secreted with VLDL[18], were occasionally found in nucleoplasmic LDs (Fig. 3a). Moreover, CCTα and perilipin-3, which bind to nucleoplasmic LDs[6], showed colocalization with ApoE (Fig. 3b, Supplementary

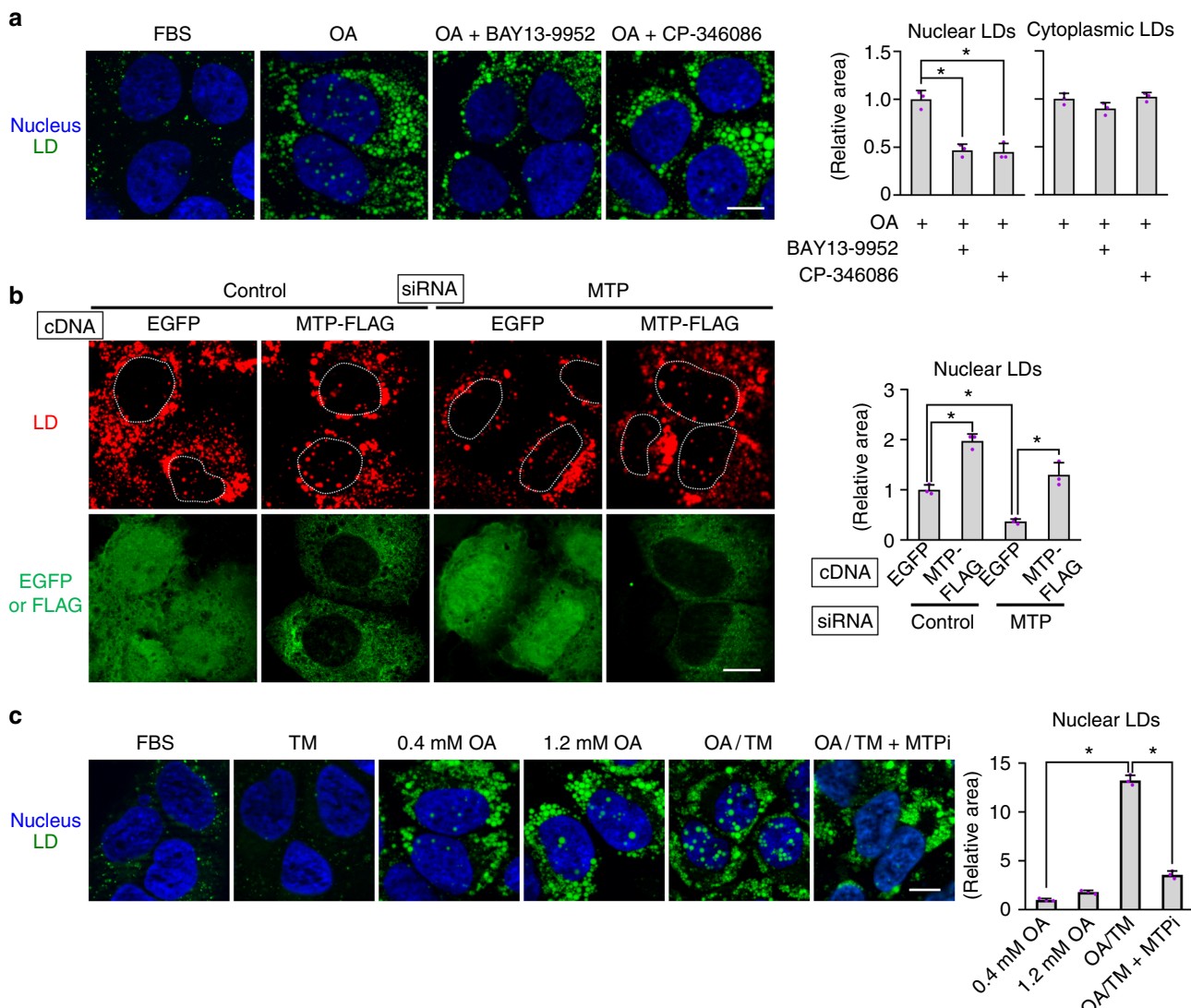

**Fig. 1** Downregulation of MTP suppresses nuclear LD formation. **a** MTP inhibitor (MTPi) suppressed the increase of nuclear LDs, but not that of cytoplasmic LDs. Huh7 cells were treated for 24 h with 0.4 mM OA with or without an MTPi, 100 nM BAY 13-9952, or 100 nM CP-346086. LD (green), nucleus (blue). **b** MTP knockdown suppressed nuclear LD formation. Huh7 cells were treated with either control or MTP siRNA, transfected with cDNA of either EGFP or MTP-FLAG, and cultured with 0.4 mM OA for 24 h. MTP-FLAG not only canceled the effect of MTP knockdown, but also increased the nuclear LDs significantly over the control level. See also Supplementary Fig. 1b. **c** OA/TM (0.4 mM OA and 5 μg/ml TM) induced a drastic increase of nuclear LDs in Huh7, and the increase was suppressed by MTPi. Although treatment with 1.2 mM OA caused an increase of nuclear LDs, treatment with TM (5 μg/ml) alone did not. **a–c** Mean ± SD of three independent experiments. *$p < 0.01$, one-way ANOVA followed by Tukey's test. Bars, 10 μm. Source data are provided as a Source data file

Fig. 3a). The result implied that NR-lumenal LDs may become exposed to the nucleoplasm at least partially through some defect in the NR membrane.

Live imaging revealed that the LBR ring around the NR-lumenal LD was disintegrated to relocate the entire LD to the nucleoplasm (Fig. 3c, Supplementary Movie 1). Defects in the NR membrane around the NR-lumenal LD were also observed by EM (Fig. 3d). Moreover, the contents of NR-lumenal and nucleoplasmic LDs often exhibited continuity through the NR membrane defect (Fig. 3e, Supplementary Fig. 3b). The latter structure was observed by immunofluorescence microscopy as ApoE-positive NR-lumenal LDs bound with smaller nucleoplasmic LDs labeled for CCTα and perilipin-3 (Fig. 3b, Supplementary Fig. 3a). These results indicated that NR-lumenal LDs turn into nucleoplasmic LDs by disintegration of the surrounding NR membrane, and that LDs newly exposed to the nucleoplasm might fuse with existing nucleoplasmic LDs.

To study whether and how NR-lumenal LDs and nucleoplasmic LDs grow in situ, LDs metabolically labeled with BODIPY-$C_{12}$ were bleached with a laser and the fluorescence recovery was examined. Fluorescence recovery was observed in both kinds of LDs, indicating that they both grow in situ (Fig. 3f, Supplementary Fig. 3c). Triacsin C retarded fluorescence recovery in both LDs, whereas MTPi affected only NR-lumenal LDs (Fig. 3f). Based on this FRAP data along with the result that prolonged MTP inhibition reduces both NR-lumenal LDs and nucleoplasmic LDs (Figs. 1a, 1c, 2f), we concluded that NR-lumenal LDs grow continuously through MTP-dependent lipid transfer, whereas nucleoplasmic LDs derive from NR-lumenal LDs, but once formed, grow largely by a mechanism independent of MTP. Diacylglycerol acyltransferase 2 in the NR may be involved in the growth of nucleoplasmic LDs[6]. Overall, the mechanism of nuclear LD formation in hepatocytes is drastically different from that reported in yeast[19].

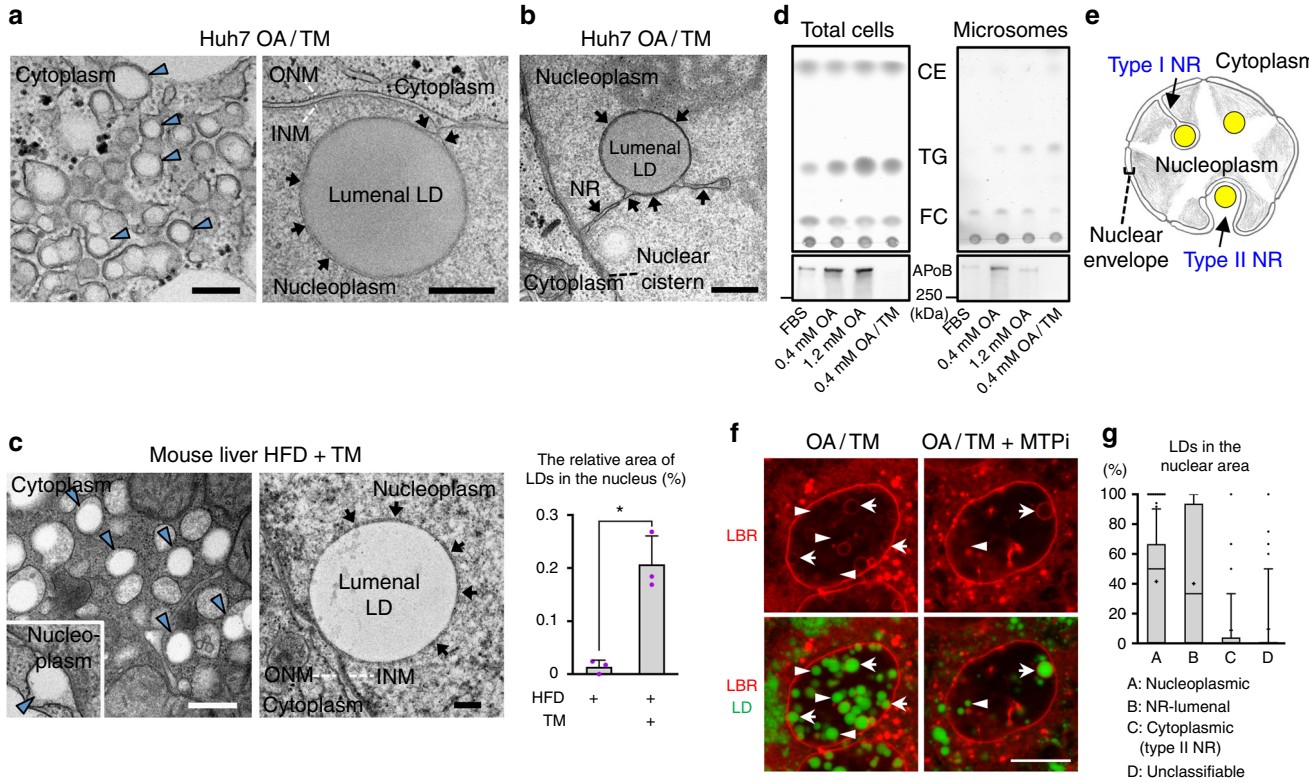

**Fig. 2** LDs are present in the lumen of the type I NR. **a** Huh7 treated with OA/TM for 24–48 h harbored LDs in the lumen of the ER (left; arrowheads) and within the type I NR (right; arrows mark the NR). INM: inner nuclear membrane, ONM: outer nuclear membrane. Bars, 0.2 μm. **b** Huh7-expressing HRP-KDEL treated with OA/TM for 24 h. DAB precipitated in the type I NR lumen (arrows). Bar, 0.5 μm. **c** Mouse hepatocytes in vivo after high-fat diet feeding for 6 weeks and TM injection. Lumenal LDs were observed in the ER (arrowheads), the nuclear cistern (arrowhead in the inset) (left figure; Bar, 0.5 μm), and in the type I NR (arrows mark the NR) (right figure; Bar, 0.2 μm). They contained more nuclear LDs than the control fed the high-fat diet and injected with vehicle alone. Mean ± SD of three independent experiments. *p < 0.01, Student's t test. **d** Microsomes of Huh7 treated with none, 0.4 mM OA, 1.2 mM OA, or OA/TM for 48 h. The OA/TM-treated cell microsome contained triglycerides (TG) and cholesterol esters (CE) most abundantly (by thin layer chromatography), but showed the lowest amount of ApoB (by Western blotting). **e** Three different kinds of LDs in the nuclear area: Nucleoplasmic LDs **(A)**, NR-lumenal LDs (within the type I NR) **(B)**, and cytoplasmic LDs (within the type II NR) **(C)**. **f** Nucleoplasmic LDs (arrowheads) and NR-lumenal LDs (arrows) are distinguished by whether they are outside of or within LBR rings, respectively. Huh7 treated with OA/TM for 48 h. Both LDs were reduced by MTPi (100 nM BAY 13-9952). Bar, 10 μm. See also Supplementary Fig. 2e. **g** The number of nucleoplasmic LDs, NR-lumenal LDs, and cytoplasmic LDs within the type II NR were counted in randomly taken electron micrographs of Huh7 treated with OA/TM for 48–72 h. Box plot of pooled data from three independent experiments. The average is shown by +. Number of nuclei examined = 128. Source data are provided as a Source data file

**NR development is separable from nuclear LD formation**. We previously showed that PML-II knockdown decreases both nuclear LDs and the NR in cells treated with OA[6]. This result was reproduced in cells treated with OA/TM (Supplementary Fig. 4a). In contrast, MTPi suppressed the increase of nuclear LDs, but not that of the NR (Fig. 4a). Furthermore, TM alone increased the NR in Huh7 without affecting nuclear LDs (Figs. 1c, 4a), whereas neither OA/TM nor TM alone increased the NR in non-hepatocytes (Supplementary Fig. 4b). These results indicated that the NR development under ER stress is unique to hepatocytes and is driven by a mechanism independent from nuclear LD formation.

Consistently, when OA/TM-treated Huh7 was transferred to a fresh medium without OA/TM, the NR decreased, whereas nuclear LDs persisted, causing a quick increase of nucleoplasmic LDs (Fig. 4b). Relocation of NR-lumenal LDs to the nucleoplasm was also observed (Supplementary Movie 2). In this process, GFP-tagged perilipin-3 was recruited to the NR-lumenal LD immediately after disruption of the NR ring, indicating that the lumenal LD becomes accessible to nucleoplasmic proteins only when the NR membrane is disintegrated (Supplementary Movie 3). These changes occurred even when MTPi was added

to the fresh medium, indicating that MTP is not necessary in this process (Supplementary Fig. 4c).

Notably, lamins were deficient in the LBR ring surrounding NR-lumenal LDs, whereas they were present in the linear NR portion and along the nuclear periphery (Fig. 4c). A similar lamin deficiency was observed in the PML-II patch of the INM[6], suggesting PML-II might be involved in the NR disintegration.

The results so far suggested that OA with or without TM increases NR-lumenal LDs in the MTP-dependent manner on the one hand and induces NR development in the PML-II-dependent manner on the other hand, and that concurrence of the two events leads to nucleoplasmic LD formation in hepatocytes.

**Nucleoplasmic LDs regenerate by the MTP-dependent process**. To examine the relationship between nuclear LDs and the NR further, mitotic cells were observed by live imaging. Interestingly, even when mother cells had abundant nuclear LDs, daughter cells immediately after mitosis had few, and gradually regained them thereafter. Several hours after mitosis, the number of nuclear LDs reached a level comparable to that of the mother cell (Fig. 5a, Supplementary Movie 4). In contrast, the NR was already present

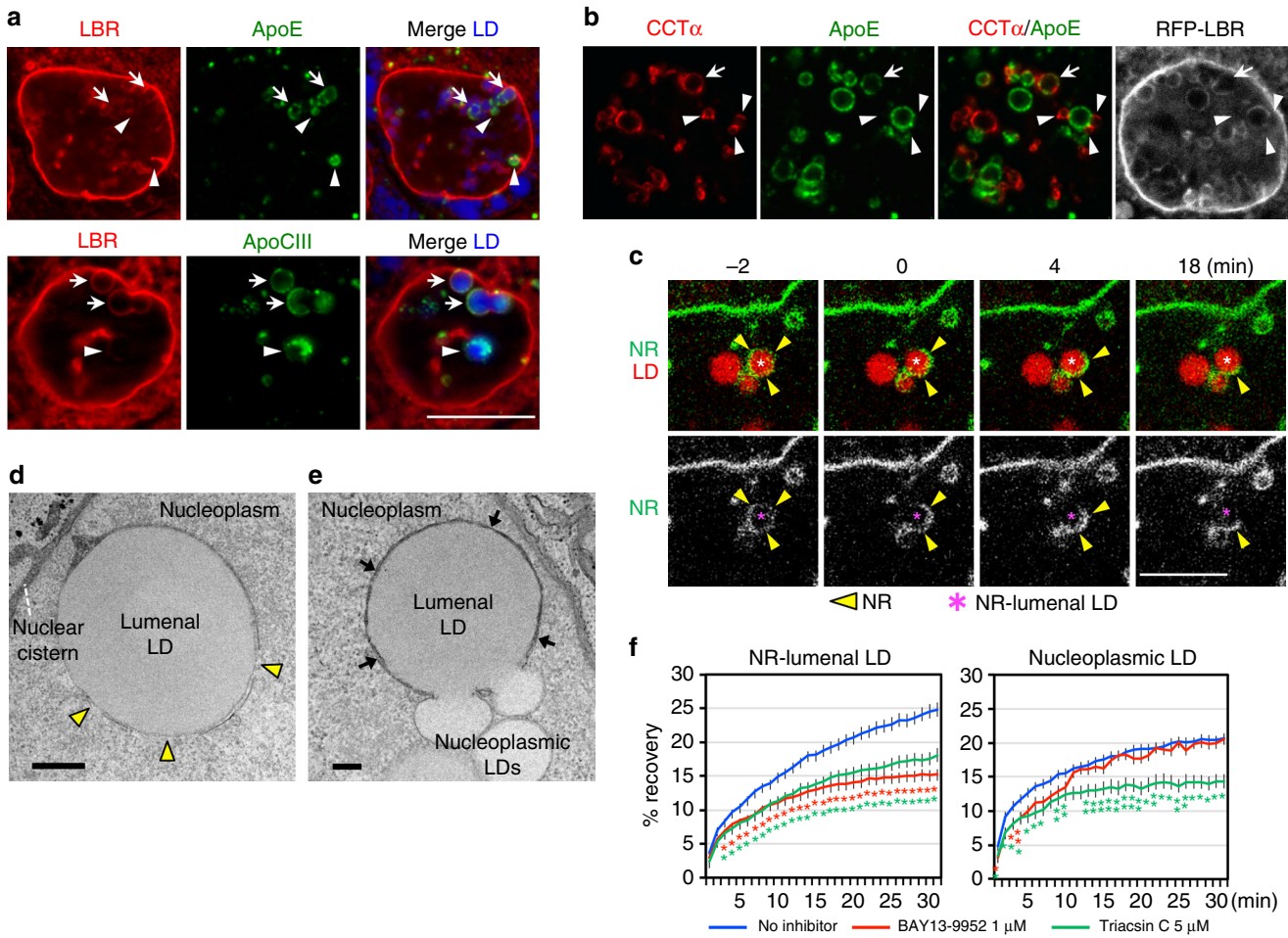

**Fig. 3** Nucleoplasmic LDs derive from NR-lumenal LDs. **a** ApoE and ApoCIII (green) were labeled in NR-lumenal LDs (arrows) and in some nucleoplasmic LDs (arrowheads). RFP-LBR (red), LDs (blue). Bar, 10 μm. Huh7 treated with OA/TM for 48 h is shown in this and subsequent figures. **b** CCTα (red) colocalized with ApoE (green) in the same LD (arrow) or distributed in small nucleoplasmic LDs (arrowheads) binding to ApoE-positive NR-lumenal LDs. The LBR ring is not distinct around the LD showing CCTα–ApoE colocalization. Bar, 10 μm. **c** Live imaging showed that the complete LBR ring (green) around an NR-lumenal LD (red) disintegrates and releases the LD into the nucleoplasm. Bar, 10 μm. Selected frames from Supplementary Movie 1. **d** The type I NR membrane around an NR-lumenal LD shows defects (arrowheads). HRP-KDEL was used to delineate the NR lumen by DAB precipitates. Bar, 0.5 μm. **e** Coalescence between NR-lumenal and nucleoplasmic LDs through defects in the NR membrane. Arrows indicate the NR. Bar, 0.2 μm. See Supplementary Fig. 3b for serial sections. **f** FRAP of LDs labeled with BODIPY 558/568-C$_{12}$. Fluorescence recovery in NR-lumenal LDs was retarded by both triacsin C and BAY 13-9952, whereas that in nucleoplasmic LDs was affected only by triacsin C. Nucleoplasmic LDs clearly separated from LBR rings were chosen. The number of LDs examined: 41 (control), 44 (triacsin C), 40 (BAY 13-9952) (NR-lumenal LDs); 36 (control), 36 (triacsin C), 21 (BAY 13-9952) (nucleoplasmic LDs). Mean ± SEM. *$p < 0.01$, **$p < 0.05$, one-way ANOVA followed by Tukey test. See also Supplementary Fig. 3c. Source data are provided as a Source data file

in daughter cells immediately after mitosis, and its number did not change before and after mitosis (Fig. 5a). It is notable that nuclear LDs appearing after mitosis codistributed with the NR initially, indicating that they are NR-lumenal LDs (Supplementary Fig. 5a). Consistently, the formation of nuclear LDs in daughter cells was suppressed by MTPi (Fig. 5b, Supplementary Movie 5).

To specifically observe nucleoplasmic LDs, we created a fluorescent marker protein, EGFP-NLSx3-HPos, by tagging three tandem nuclear localization sequences (NLS) to HPos[20] (Supplementary Fig. 5b, c). By using EGFP-NLSx3-HPos, we confirmed that nucleoplasmic LDs decreased in the anaphase and increased gradually after mitosis, and that MTPi suppressed the post-mitotic recovery (Fig. 5c, Supplementary Movie 6). These results corroborated the idea that the NR is present immediately after mitosis, whereas nucleoplasmic LDs disappear during mitosis and form anew in daughter cells in the MTP-dependent manner.

**CCTα recruited to nucleoplasmic LDs activates PC synthesis**. To study the functional significance of nucleoplasmic LDs, we focused on CCTα that was labeled in nucleoplasmic LDs in hepatocarcinoma cell lines[6] and in primary cultured mouse hepatocytes (Supplementary Fig. 6a). MTP inhibition, which suppressed nucleoplasmic LD formation, drastically decreased PC synthesis without affecting the expression of CCTα and other Kennedy pathway enzymes (Supplementary Fig. 6b). The result led us to hypothesize that nucleoplasmic LDs may be a major site of CCTα activation in hepatocytes.

We noticed that most nucleoplasmic LDs were predominantly labeled for either CCTα or perilipin-3 and rarely for both (Fig. 6a, Supplementary Fig. 6c). Considering that perilipins are more resistant to the protein crowding effect than amphipathic proteins like CCTα[21], the mutually exclusive distribution suggested that perilipin-3 may work as a dominant factor to regulate the binding of CCTα to nucleoplasmic LDs. In support of this idea, perilipin-

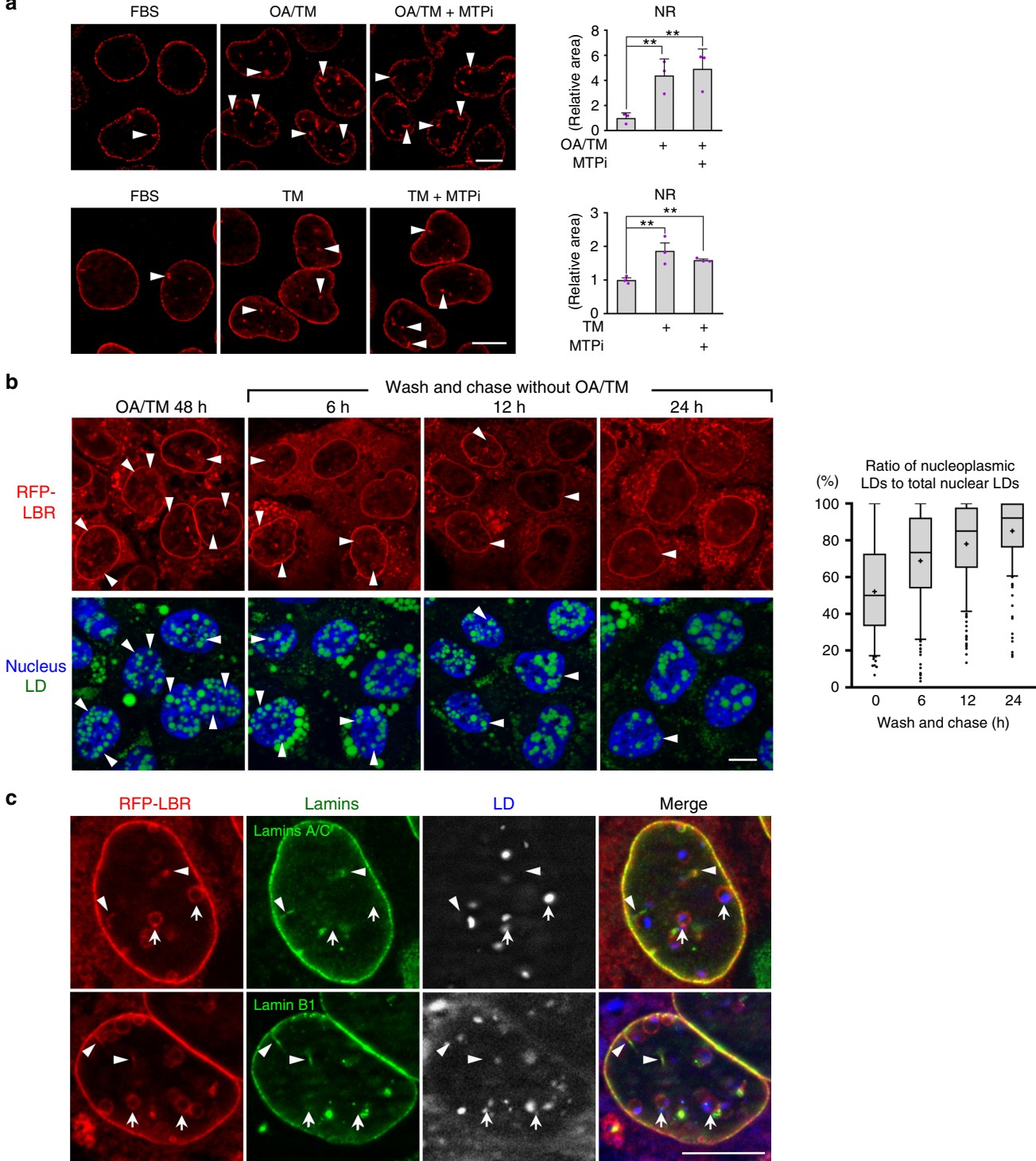

**Fig. 4** MTP inhibition does not affect the development of NR. **a** The NR (red, endogenous LBR; arrowheads) increased in Huh7 treated for 48 h with OA/TM (upper panel) or TM alone (lower panel), and the increase was not suppressed by MTPi (100 nM BAY 13-9952). Mean ± SD of three independent experiments. **p < 0.05, one-way ANOVA followed by Tukey's test. See also Supplementary Fig. 4b. **b** The increase of nucleoplasmic LDs after OA/TM washout. Huh7 pretreated with OA/TM for 48 h was chased in a fresh medium. Shrinkage of the NR (red, RFP-LBR; arrowheads) caused an increase of nucleoplasmic LDs and a decrease of NR-lumenal LDs. LDs (green), nucleus (blue). Box plot of pooled data from three independent experiments. The average is shown by +. Number of nuclei counted: 194 (0 h), 187 (6 h), 209 (12 h), and 172 (24 h). **c** Deficiency of lamins in the LBR ring. Lamin A/C and lamin B1 (green) were labeled in the linear NR portion (arrowheads) and along the nuclear periphery, but not in the ring-shaped NR around NR-lumenal LDs (arrows). Huh7 treated with OA/TM for 48 h. RFP-LBR (red). **a–c** Bars, 10 µm. Source data are provided as a Source data file

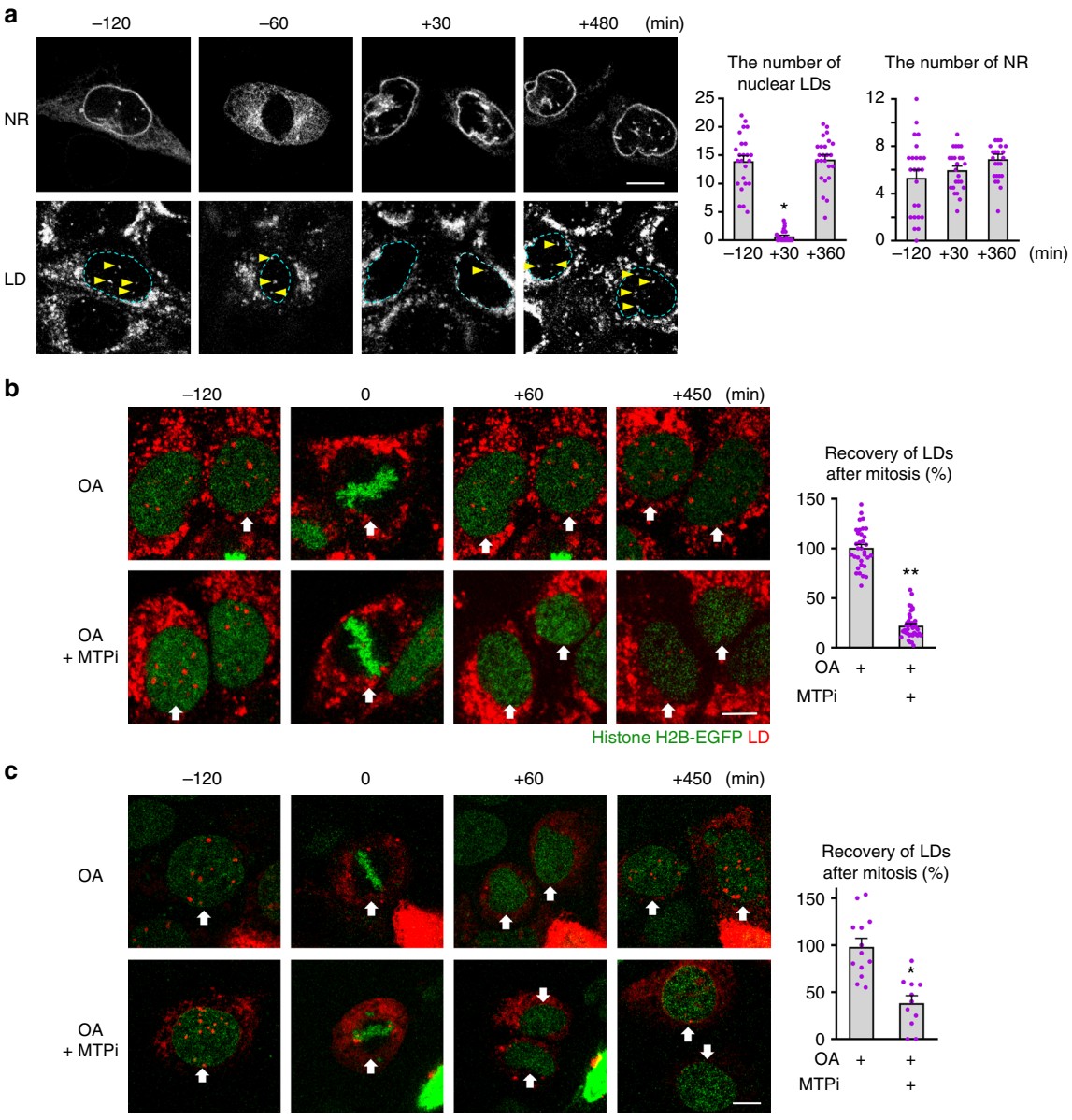

**Fig. 5** Nuclear LDs regenerate in post-mitotic cells in an MTP-dependent manner. **a** Huh7 preincubated with OA for 24 h was tracked in the same medium. Selected frames from Supplementary Movie 4. LDs (LipidTox Red, arrows), INM and NR (EGFP-LBR). Nuclear LDs and the NR were counted 2 h before mitosis, and 0.5 h and 6 h after mitosis (0 min = the time point of nuclear envelope reformation). Mean ± SEM of 24 mitotic events. *$p < 0.01$, Friedman test followed by Dunn–Bonferroni test. The lack of difference in the NR was verified by repeated measures ANOVA. **b** Huh7 preincubated with OA for 24 h was tracked in the same medium with or without 100 nM BAY 13-9952. Selected frames from Supplementary Movie 5. LDs (red, LipidTox Red), chromatin (green, histone H2B-EGFP). The number of nuclear LDs before mitosis and 6 h after mitosis was counted (0 min = the time point that cells of interest entered the metaphase). Mean ± SEM of 33 (OA) and 40 (OA and BAY 13-9952) mitotic events. *$p < 0.01$, Student's $t$ test. **c** Huh7 preincubated with OA for 24 h was tracked in the same medium with or without 1 μm BAY 13-9952. Selected frames from Supplementary Movie 6. Nucleoplasmic LDs (red, EGFP-NLSx3-HPoS), chromatin (green, histone H2B-mCherry). Nuclear LDs before mitosis and 6 h after mitosis was counted. Mean ± SEM of 13 (OA) and 11 (OA and BAY 13-9952) mitotic events. *$p < 0.01$, Student's $t$ test. Bars, 10 μm. Source data are provided as a Source data file

3 knockdown increased CCTα-positive nucleoplasmic LDs (Fig. 6b, Supplementary Fig. 6d). Moreover, perilipin-3 knockdown increased PC synthesis (Fig. 6c, Supplementary Fig. 6d) without upregulating expression of PC synthetic enzymes (Supplementary Fig. 6e). The result indicated that an increase in CCTα recruitment to nucleoplasmic LDs activates PC synthesis.

The effect of perilipin-3 knockdown on PC synthesis, however, might be caused indirectly by an unknown mechanism through a decrease of perilipin-3 in the cytoplasm. To examine this

possibility, we constructed perilipin-3 appended with a nuclear exclusion signal (perilipin-3-NES), which distributed to cytoplasmic LDs but not to nuclear LDs (Fig. 6d). Whereas expression of the wild-type perilipin-3 significantly decreased CCTα in nucleoplasmic LDs, expression of perilipin-3-NES had no effect (Fig. 6d). The result indicated that CCTα in nucleoplasmic LDs is directly affected by perilipin-3 binding to nucleoplasmic LDs.

To examine the effect of perilipin-3 overexpression on PC synthesis, we used an imaging method employing a clickable choline analog[22], because a low cDNA transfection efficiency of

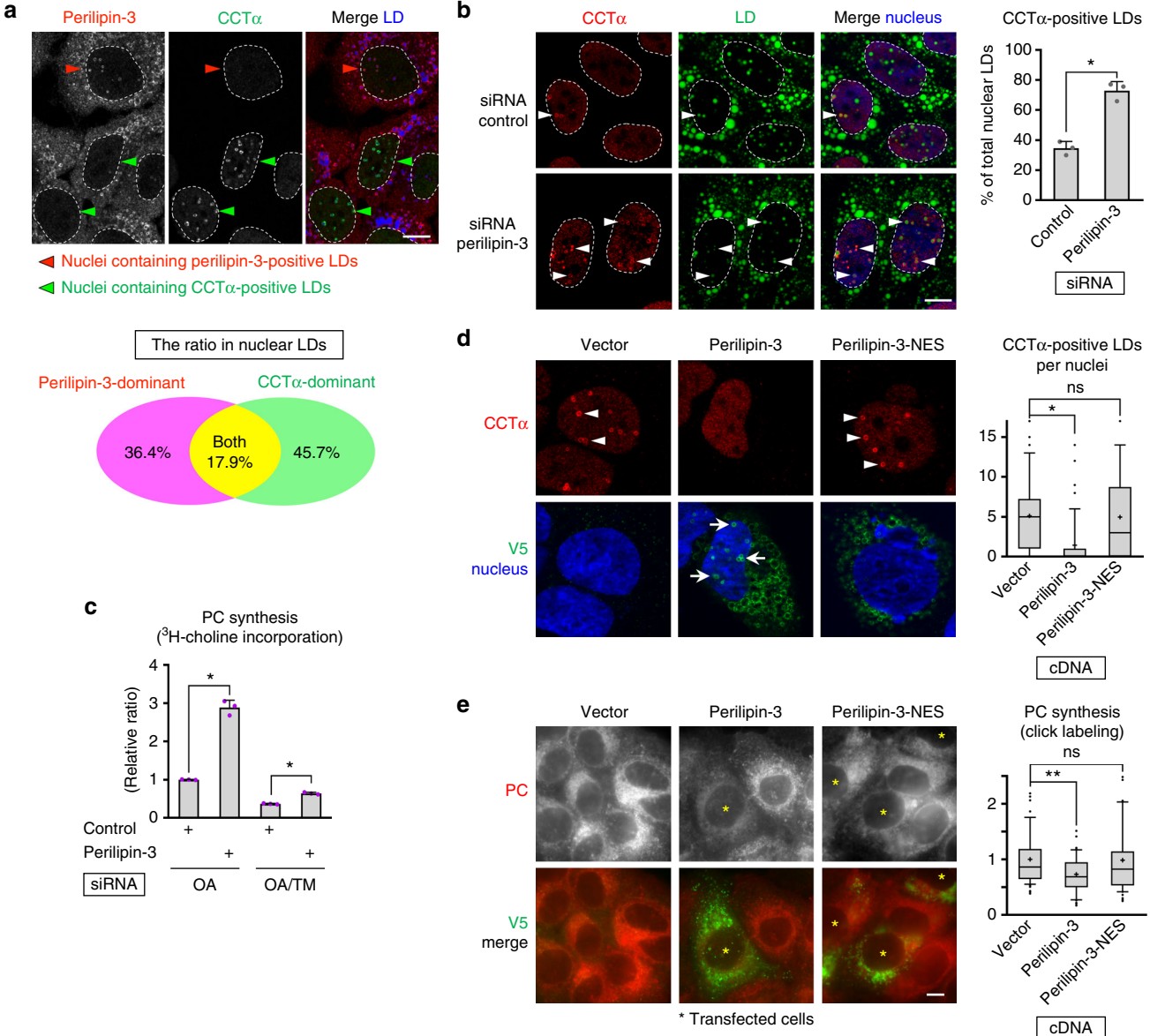

**Fig. 6** CCTα recruited to nucleoplasmic LDs activates PC synthesis. **a** Either perilipin-3 (red) or CCTα (green) is predominant in a majority of nuclear LDs (blue). Huh7 treated with OA for 24 h. The LDs (total number, 396) were classified into perilipin-3-dominant, CCTα-dominant, and the other (see Supplementary Fig. 6c); the Venn diagram shows the proportion of the three groups. **b** Knockdown of perilipin-3 increased nuclear LDs harboring CCTα (arrowheads). Huh7 transfected with either control siRNA or perilipin-3 siRNA #1 was treated with OA for 24 h. CCTα (red), LDs (green), and nuclei (blue). Mean ± SD of three independent experiments. *$p < 0.01$, Student's $t$ test. **c** Knockdown of perilipin-3 increased PC synthesis. $^{3}$H-choline incorporation to Huh7 treated as in Fig. 6b was measured. Mean ± SD of triplicate samples. *$p < 0.01$, Student's $t$ test. **d** Perilipin-3-V5, but not perilipin-3-NES-V5, decreased nuclear LDs harboring CCTα (arrowheads). Huh7 transfected with respective cDNA was treated with OA for 24 h. CCTα (red), V5 (green), nuclei (blue). Box plot of data from three independent experiments. The average is shown by +. The number of cells counted = 226 (empty vector), 74 (perilipin-3-V5), and 72 (perilipin-3-NES-V5). *$p < 0.01$, Kruskal–Wallis ANOVA followed by Dunn–Bonferroni test. **e** Perilipin-3-V5, but not perilipin-3-NES-V5, decreased PC synthesis. Huh7 treated as in Fig. 6d was incubated with 0.25 mm propargylcholine for 1 h and labeled for PC by a click reaction using Cy3-azide. The fluorescence intensity of Cy3 in individual cells was measured. Box plot of pooled data from three independent experiments. The average is shown by +. The number of samples counted = 57 (empty vector), 52 (perilipin-3-V5), and 50 (perilipin-3-NES-V5). **$p < 0.05$, Kruskal–Wallis ANOVA followed by Dunn–Bonferroni test. See also Supplementary Fig. 6d. Bars, 10 μm. Source data are provided as a Source data file

Huh7 precluded the use of biochemical analyses (Supplementary Fig. 7a). By using the method, PC synthesis was found to decrease in cells expressing wild-type perilipin-3, but not in those expressing perilipin-3-NES (Fig. 6e), even though the expression of perilipin-3-NES was significantly higher than that of wild-type perilipin-3 (Supplementary Fig. 7b). Furthermore, the difference in PC synthesis was not observed when CCTα was knocked down by RNAi, indicating that the result of perilipin-3 expression was caused through differential effects on CCTα activation

(Supplementary Fig. 7c). These results corroborated that nucleoplasmic LDs are a major site of CCTα activation and that perilipin-3 downregulates PC synthesis by displacing CCTα from nucleoplasmic LDs (Fig. 7).

## Discussion

We showed that nuclear LDs in hepatocytes derive from ApoB-free lumenal LDs, a precursor of VLDL[7–9]. As reported in

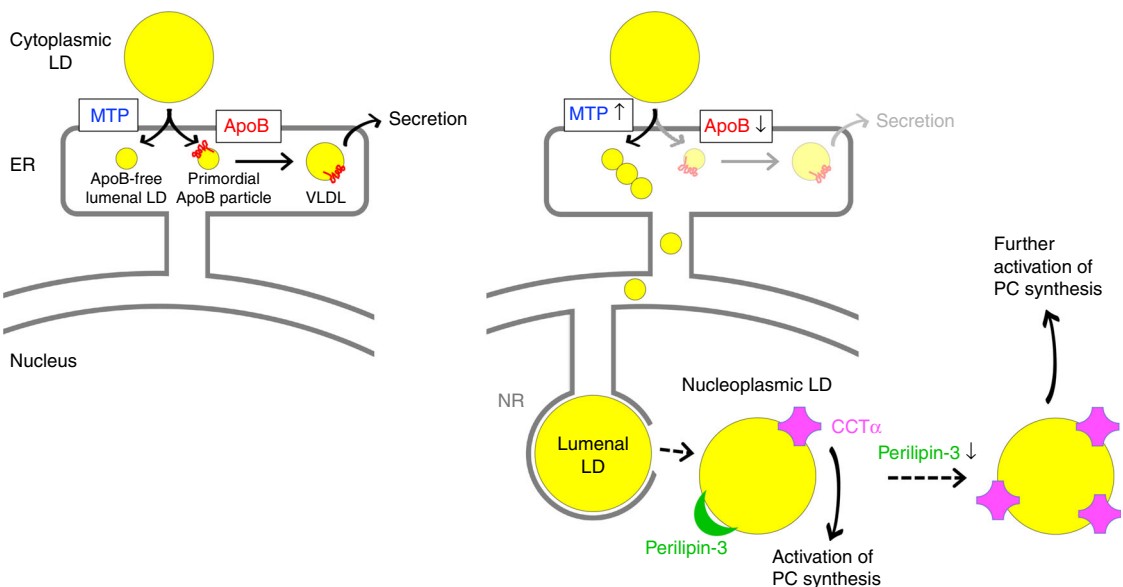

**Fig. 7** Formation and function of nuclear LDs. MTP-driven ApoB-free lumenal LD formation increases upon ER stress, resulting in lumenal LDs in the type I NR, which are relocated to the nucleoplasm through defective NR membrane. CCTα recruited to nucleoplasmic LDs activates PC synthesis, which probably contributes to mitigation of ER stress and maintenance of VLDL secretion. Downregulation of perilipin-3 enhances this mechanism by increasing the CCTα recruitment

enterocytes, when ApoB is deficient while MTP remains active, ApoB-free lumenal LDs accumulate in the ER lumen because their export to post-ER compartments is suppressed[23]. A similar condition occurs in hepatocytes under ER stress, in which ApoB is decreased through cotranslational and posttranslational degradation[24], whereas the expression of MTP is maintained[16]. ER stress enhances MTP activity further, because the inositol-requiring transmembrane kinase/endoribonuclease 1α (IRE1α)-X box binding protein 1 (XBP1) pathway increases expression of protein disulfide isomerase, the obligate cofactor of MTP[25]. In cells treated with OA/TM, these changes must combine to cause a marked accumulation of lumenal LDs. The moderate but significant increase in lumenal LDs in OA-treated cell is thought to arise by the same mechanism, because OA causes ER stress through activation of ApoB synthesis[17,26].

The type I NR also develops in hepatocytes treated with OA or OA/TM. This appears to occur independently from lumenal LD formation, because it requires PML-II, but not MTP. The requirement of the two independent factors, namely, MTP-dependent lumenal LD formation and PML-II-dependent type I NR development, is met in hepatocytes but not in most other cell types[6,7] and thus accounts for the prominent abundance of nuclear LDs in hepatocytes.

Lumenal LDs grown in the type I NR relocate to the nucleoplasm. This surprising movement is in most cases probably made possible by disintegration of the NR membrane surrounding NR-lumenal LDs. Deficiency of lamins that occurs only in that portion of the NR probably makes the membrane mechanically unstable, thus being a prerequisite for the phenomenon. Continuous growth of NR-lumenal LDs by MTP-dependent lipid transfer is thought to impose pressure to the lamin-deficient NR membrane, leading to its disruption. The mechanism causing the local lamin depletion is not clear, but a similar lack of lamins was observed in the PML-II patch region[6,27], suggesting the importance of PML-II.

The nuclear envelope rupture accompanied by local lamin depletion is known to occur in other conditions, but they invariably involve rupture of both INM and ONM[28,29]. The present finding indicates that disruption of the INM alone occurs,

not as a result of cellular damage, but in a regulated manner to maintain cellular homeostasis. In this context, it is notable that ApoE is present in the nucleus of non-hepatocytes and may be related to gene transcription[30,31]. Furthermore, simian virus 40 was proposed to enter from the ER lumen to the nucleoplasm[32]. These results suggest that the ER lumen-to-nucleoplasm route might be utilized widely and be exploited by pathogens.

The nucleoplasmic LD in hepatocytes was found to be a major site of CCTα activation. This is different from what was suggested previously[33–37]. The discrepancy may be caused by difference in frequency of nucleoplasmic LDs. But it is notable that lumenal LDs in hepatocytes contain abundant phosphatidylethanolamine (PE)[38] and thus, nucleoplasmic LDs, derived from lumenal LDs, are also likely to be rich in PE. This property is similar to that of cytoplasmic LDs in *Drosophila*, in which abundant PE is thought to facilitate the formation of packing defects due to its cone shape, causing efficient recruitment of CCT1.

We showed that upregulation of PC synthesis in ER stress occurs by activation of CCTα on the increased nucleoplasmic LDs. Enforced expression of XBP1s and a constitutively active form of ATF6α were shown to upregulate PC synthesis through increased expression of CCTα and other PC synthetic enzymes[39,40], but whether these UPR proteins function similarly at the endogenous level is not known. We speculate that the nucleoplasmic LD-based mechanism, to which MTP activation through the IRE1α-XBP1 pathway may contribute[25], has a major role in increasing PC synthesis, which is thought to mitigates ER stress by expanding the ER lumen[41].

Active PC synthesis is necessary for VLDL secretion in hepatocytes[42]. An increased supply of fatty acids stimulates ApoB synthesis, but also elicits ER stress, probably through the increase of ApoB in the ER[26], making VLDL secretion less than maximal[17,43]. Here, PC is the only phospholipid required for VLDL assembly[42] and, as such, synthesis of ApoB and PC needs to be coordinated to maintain an optimal condition. In this context, it is physiologically relevant that PC synthesis is upregulated upon increase of ApoB-free lumenal LDs, which are normally exported to Golgi but accumulate in the ER when stress mounts. Conversely, upon MTP inhibition, the resultant decrease

of ApoB-free lumenal LDs leads to suppression of PC synthesis through downregulation of the nucleoplasmic LD formation. In both cases, ApoB-free lumenal LDs are thought to be utilized as the indicator for hepatocytes to adjust the rate of PC synthesis in accordance with that of ApoB synthesis.

Physiological function of perilipin-3 has been elusive[44], but the present result indicated that perilipin-3 regulates CCTα recruitment to nucleoplasmic LDs. In this context, it is noteworthy that exposure of primary rat hepatocytes to fatty acids, which should increase VLDL synthesis, decreases perilipin-3 mRNA[45], suggesting that transcription of perilipin-3 may be regulated to adapt to PC demand. On the other hand, knockdown of perilipin-3 in mouse liver was shown to reduce hepatic steatosis induced by a high-fat diet[46]. Our result explained the mechanism behind this observation and indicated that perilipin-3 is a potential drug target to treat fatty liver disease. Further studies in this direction are warranted.

## Methods

**Antibodies**. Rabbit anti-human perilipin-3 antibody was raised against a peptide of human perilipin-3 segment (amino acids 305–318)[47]. Rabbit anti-CCTα antibody was donated by Dr. Neale Ridgway (Dalhousie University). Rabbit anti-lamin B receptor (Genway Biotech GWB-C7CA28), rabbit anti-lamin B1 (Abcam ab16048), mouse anti-lamin A/C (Cell Signaling #4777), rabbit anti-actin (Sigma A2066), and mouse anti-FLAG (Sigma F1804), goat anti-MTP (Santa Cruz sc-33116), mouse anti-ApoE (Immunogenetics M-012-0500), mouse anti-choline kinase β (Santa Cruz sc-398957), mouse anti-diacylglycerol cholinetransferase 1 (Santa Cruz sc-515577), rabbit anti-choline/ethanolamine phosphotransferase 1 (Abgent #AP10372a), goat anti-ApoB (Rockland 600-101-111), rabbit anti-ApoC-III (Thermo Fisher 6H21L11), mouse anti-V5 (Thermo Fisher R960-25), mouse anti-choline kinase α (CKα) (Proteintech 13520-1-AP), guinea pig anti-perilipin-3 (Progen GP30), and mouse anti-nucleopore complex (Covance MMS-120P) antibodies were obtained from the respective suppliers. Secondary antibodies conjugated to fluorochromes and peroxidase were purchased from Thermo Fisher, Jackson ImmunoResearch Lab, and Bethyl Laboratories. See Supplementary Table 1 for dilutions of antibodies used for immunofluorescence labeling and Western blotting.

**Cell lines and reagents**. HepG2 (JCRB1054), HEK293 (JCRB9068), and HeLa (JCRB9004), which were obtained from the Japanese Collection of Research Bioresources Cell Bank, McA-RH7777 (ATCC CRL-1601) obtained from American Type Culture Collection, and A549 donated by Dr. Takashi Takahashi (Nagoya University) were cultured in Dulbecco's modification of Eagle medium (DMEM) supplemented with 10% fetal bovine serum (FBS) and antibiotics at 37 °C in a humidified atmosphere of 95% air and 5% $CO_2$. Huh7 from Dr. Eija Jokitalo (University of Helsinki) and U2OS from Dr. Hidemasa Goto (Aichi Cancer Center) were cultured in MEM and McCoy's medium, respectively, with the same supplementations. In some experiments, OA (Sigma) in complex with fatty acid-free bovine serum albumin (BSA) (Wako) at a molar ratio of 6:1 was added. BAY 13-9952 (Implitapide; Bayer Healthcare), triacsin C (Santa Cruz Biotech), CP-346086, and TM (Sigma) were purchased from respective suppliers.

**Primary mouse hepatocytes**. Hepatocytes were isolated from 8-week-old mice under anesthesia using the two-step collagenase perfusion method[48]. In brief, HEPES-buffered saline containing 0.5 mM EGTA (ethylene glycol-bis(β-aminoethyl ether)-N,N,N′,N′-tetraacetic acid)and then a digestion medium with collagenase (100 digestive units/ml) were perfused via the inferior vena cava, and isolated cells were plated on collagen I-coated coverslips in DMEM supplemented with 10% FBS. The cells were used within 2–3 days after isolation.

**Transfection of plasmid and siRNA**. pEGFP (Takara Bio), pCMV-Tag4 (Agilent), and pcDNA3.1/TOPO-V5 (Thermo Fisher) were purchased from respective suppliers. Plasmids for HRP-KDEL, EGFP-KDEL, mRFP-LBR, and EGFP-LBR were donated by Dr. Colin Hopkins (University College London), Dr. Katsuhiko Mikoshiba (Riken), and Dr. Tokuko Haraguchi (Advanced ICT Research Institute), respectively. Plasmids for histone H2B-mCherry and histone H2B-EGFP were obtained from Addgene. Plasmids for HPos[20] and LiveDrop[49] were constructed according to respective reports. Nuclear localization signal (NLS), KRPAATK-KAGQAKKKK, and nuclear export signal (NES), LQLPPLERLTLD, were taken from Xenopus nucleoplasmin (Accession No. X04766.1) and HIV Rev[50], respectively. Some plasmids were cloned from cDNA of Huh7 by reverse transcription polymerase chain reaction (RT-PCR) using TRIZOL and Superscript III reverse transcriptase (Thermo Fisher). Human perilipin-3 cloned to pcDNA3.1/TOPO-V5 (Thermo Fisher)[47] was added with NES to make perilipin-3-NES-V5. See information of primers and oligonucleotides in Supplementary Table 2.

siRNAs were synthesized by Japan Bio Service, Inc. Target nucleotide sequences of siRNAs were adopted from previous publications. For MTP, a cocktail of the following three siRNAs were used:[51] 5'-GGAAAUGCCUGCAAGCAAATT-3' (sense), 5'-UUU GCUUGCAGGCAUUUCCTT-3' (antisense); 5'-GUCUAAAACCCGAUGAAATT-3' (sense), 5'-UUUCACUCGGGUUUUUAGACTT-3' (antisense); 5'-GGUAGAAGGCA CAUAGAAATT-3' (sense), 5'-UUUCUAUGUGCCUUCUACC-3' (antisense). Sequences of other siRNAs are as follows: for perilipin-3 #1[47], 5'-GGAACAGAG CUACUUCGUATT-3' (sense), 5'-UACGAAGUAGCUCUGUUCCTT-3' (antisense); for perilipin-3 #2[52], 5'-GUUCUCCCCUGCAGAAUUUTT-3' (sense), 5'-AAAUUCUG CAGGGGAGAACTT-3' (antisense); and for PML-II[6], 5'-CAUCCUGCCCAGCUGC AAATT-3' (sense), 5'-UUUGCAGCUGGCAGGAUGTT-3' (antisense). For perilipin-3 knockdown, siRNA #1 was used, except in the experiments shown in Supplementary Fig. 6d.

cDNA was transfected using Lipofectamine 2000 or Lipofectamine 3000, and RNA was transfected using Lipofectamine RNAiMAX (Invitrogen) according to the manufacturer's instruction. The cells were analyzed two days (cDNA) and three days (siRNA) after transfection.

**Immunofluorescence microscopy and data analysis**. Cells were fixed with 3% formaldehyde alone or 3% formaldehyde and 0.015% glutaraldehyde in 0.1 M phosphate buffer for 15 min, and permeabilized with either 0.01% digitonin in phosphate-buffered saline (PBS) (room temperature, 30 min; for ApoB, ApoE, perilipin-3, and perilipin-3-V5), 90% methanol (− 20 °C, 10 min; for double labeling of CCTα and perilipin-3), 0.05% Triton X-100 in PBS (on ice, 10 min; for CCTα), 0.1 % Triton X-100 in PBS (room temperature, 10 min; for ApoCIII), or 0.5% Triton X-100 in PBS (room temperature, 10 min; for other proteins). Blocking and antibody dilution were carried out with 3% BSA and 0.01% digitonin in PBS (for ApoE, ApoB, perilipin-3, perilipin-V5, and double labeling of perilipin-3 and CCTα), 3% BSA and 0.05% Triton X-100 in PBS (for CCTα), and 1% BSA in PBS (for ApoCIII) or 3% BSA in PBS (for other proteins). Nuclei were labeled with Hoechst 33342 (Sigma), whereas LDs were labeled with either BODIPY493/503, LipidTox Red (Thermo Fisher), or Ac202 (AVICOR). Samples were mounted in Mowiol 4–88 containing 2.5% 1,4-diazabicyclo-[2,2,2]-octane.

Images of random areas were captured by either a confocal laser scanning microscope A1 (Nikon) using a PlanApo 100 × /1.45 or an Axio Imager 2 equipped with Apotome2 (Carl Zeiss) using a Plan-Neofluar ×100 /1.30. For quantification, the number of LDs was counted manually, and the area was measured using ImageJ. The relative area of nuclear LDs and the NR was obtained by dividing the total nuclear LD area and the total intranuclear LBR-positive area by the nuclear area, respectively. For the NR measurement, the signal in the nuclear periphery was excluded. More than 100 cells were examined for each group in each experiment, and results from three independent experiments were averaged unless stated otherwise. The color, brightness, and contrast of presented images were adjusted using Adobe Photoshop CS3 or Zeiss Zen.

**Click chemistry for detection of de novo synthesized PC**. Cells were cultured with 0.25 mM propargylcholine for 60 min, fixed, and subjected to click chemical conjugation with Cy3-azide (baseclick) to visualize synthesized PC[22]. The integrated fluorescence intensity of Cy3-azide labeling per cell area was quantified in randomly taken images by using ImageJ.

**Live imaging**. Live confocal images were obtained using a Cell Voyager CV1000 spinning disk confocal laser confocal system (Olympus) equipped with 488 nm and 561 nm diode lasers. An electron multiplying charge-coupled device camera (Hamamatsu Photonics. 1000 × 1000 pixels, ImagEM C9100-14 and 512 × 512 pixels, ImagEM C9100-13) and oil immersion objective lenses (Olympus; UPLSAPO ×100 /1.4 and UPLSAPO ×60 /1.35) were used. Cells were kept in a culture chamber filled with 95% air and 5% $CO_2$ and observed continuously for up to 24 h at 2 min intervals. For each point, multiple z-stack sections of < 1.0 μm intervals were obtained. Images were processed by Fiji application.

**Electron microscopy**. Cells were fixed with a mixture of 2% formaldehyde and 2.5% glutaraldehyde in 0.1 M sodium cacodylate buffer (pH 7.4), and postfixed with a mixture of 1% osmium tetroxide and 0.1% potassium ferrocyanide in 0.1 M sodium cacodylate buffer. Cells expressing HRP-KDEL were incubated in an enzyme histochemical reaction solution to form DAB precipitates before osmification[53,54]. Samples were embedded in epoxy resin, and ultrathin sections were observed using a JEOL JEM1011 electron microscope operated at 100 kV.

**Florescence recovery after photobleaching**. Huh7 cells expressing EGFP-LBR were treated with OA/TM, metabolically labeled with BODIPY 558/568-$C_{12}$ (Invitrogen) for 1 h, rinsed, and cultured for another 1 h. For some experiments, either BAY 13-9952 or triacsin C was added 15 min before photobleaching. Up to three different NR-lumenal or nucleoplasmic LDs were bleached five times with 63 ms intervals using 40% maximum intensity of the 561 nm laser. Images were captured for 30 min at 1 min intervals with a confocal laser scanning microscope (TiE-A1R, Nikon), using a Plan Apo ×100 /1.45 oil immersion lens. All experiments were performed at 37 °C in a humidified atmosphere of 95% air and 5% $CO_2$. Florescence recovery in ROIs was quantified using ImageJ and normalized to

fluorescence in the LDs before bleaching. Results from three independent experiments were averaged.

**Microsome fractionation**. All of the procedures were carried out at 4 °C. Cells homogenized in hypotonic buffer (10 mM HEPES, pH 7.4, 1.5 mM MgCl$_2$, 10 mM KCl, 2 mM dithiothreitol) were centrifuged at 218 g for 10 min, and then at 15,000 g for 10 min, to remove nuclei and mitochondria. The supernatant was centrifuged at 106,000 g for 1 h to obtain microsome as a pellet, which was re-suspended in ice-cold 20 mM Tris-HCl (pH 7.4) and 0.5 M NaCl to dissociate LDs and then centrifuged at 106,000 g for 1 h. The final pellet was used for western blotting and thin layer chromatography.

**Western blotting**. For conventional western blotting, cells were either directly dissolved in sodium dodecyl sulphate (SDS) sample buffer or treated with radio-precipation (RIPA) buffer (1% Triton X-100, 50 mM Tris-HCl, pH 7.4, 150 mM NaCl, 0.1% SDS, 0.5% sodium deoxycholate), and the supernatant was used. For detection of phosphorylated eIF2α, RIPA buffer was added with 50 mM NaF, 30 mM sodium molybdate, and 2 mM sodium orthovanadate. The reaction obtained with Super Signal West Dura Substrate (Thermo Fisher) was visualized using a Fusion Solo S instrument (Vilber Lourmat) and analyzed by Fusion-Capt Advance Software version 16.15. Uncropped western blots are shown in the Source Data file.

**Lipid extraction and thin layer chromatography**. Lipids were extracted from the total cell lysate and the microsomal fraction by using the method of Bligh and Dyer[55]. In brief, one volume of the sample was mixed sequentially with 3.75 volume of chloroform/methanol (1/2), 1.25 volume of chloroform, and 1.25 volume of distilled water. The organic phase was dried, dissolved in chloroform/methanol (1/1), and separated on HPTLC Silica gel 60 (Merck) by hexane/diethylether/acetic acid (80/20/1) together with lipid standards. Lipid spots on the plate were visualized by staining with 3% copper acetate in 85% phosphoric acid and heating at 180 °C.

**$^3$H-choline incorporation assay**. Cells cultured with 1 μCi/ml $^3$H-choline (Perkin Elmer) for 30 min were incubated with hexane/isopropanol (9/1) for 30 min to selectively extract PC among choline-containing metabolites[56]. The $^3$H-choline radioactivity was measured using a liquid scintillation counter (Aloka) and normalized to the protein concentration.

**Animal experiment**. Ten-week-old Slc:ddY male mice were fed a high-fat diet (58 kcal% fat; HFD-32, Oriental Yeast) for 6 weeks and injected intraperitoneally with TM (1 μg/g body weight in 0.15 M sucrose) or vehicle alone, one shot per day for 2 days. Anesthetized mice were fixed by a perfusion of aldehydes and the excised liver was processed for EM. The mouse experiment conforms to the Guidelines for the Proper Conduct of Animal Experiments of the Science Council of Japan and was approved by the Animal Experimentation Committee of Nagoya University Graduate School of Medicine (Approval ID: 29097).

**Statistical analysis**. Statistical differences between samples were examined by one-way analysis of variance (ANOVA) for independent groups followed by Tukey's test for Figs. 1, 3f, and 4a; Kruskal–Wallis ANOVA followed by the Dunn–Bonferroni test for Fig. 6d and 6e; Friedman test followed by Dunn–Bonferroni test and repeated measures ANOVA for Fig. 5a; Student's t test for independent groups for the other. Differences were considered significant and marked as * ($p < 0.01$) and ** ($p < 0.05$). Statistical analyses were performed with PQStat version 1.4.8 and Microsoft Excel version 16.14.1. The number of technical replicates and/or counts are described in each figure legend.

**Box plots**. For the data, in which the number of samples exceeded 50, the result was shown in box plots. They were prepared using Prism 8 (GraphPad). The center lines show the median, box boundaries indicate the 25th and 75th percentiles, whiskers delineate the 10–90 percentile range, and dots represent individual data points. The average is shown by +.

**Reporting summary**. Further information on experimental design is available in the Nature Research Reporting Summary linked to this article.

## Data availability

Data supporting the findings of this manuscript are available from the corresponding authors upon reasonable request. A reporting summary for this Article is available as a Supplementary Information file. The source data underlying Figs. 1a–c, 2c, d, g, 3f, 4a, b, 5a–c, 6a–e, and Supplementary Figs. 1a–c, 4a, 6b, d, e, 7a–c are provided as a Source Data file.

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

## Acknowledgements

We thank staff members of the Division for Medical Research Engineering and the Radioisotope Research Center of Nagoya University Graduate School of Medicine for skillful support, Dr. Takaki Miyata (Nagoya University), and Dr. Yusuke Miyanari (National Institute for Basic Biology) for generous permission to use microscopes, and many colleagues for donation of precious materials. Kamil Sołtysik is a recipient of the Japanese Government MEXT fellowship. This study was supported by JSPS KAKENHI to Yuki Ohsaki (18K06829), Toyoshi Fujimoto (15H05902, 18H04023), and Advanced Bioimaging Support (JP16H06280)

## Author contributions

K.S., Y.O. and T.F. designed the experiments and analyzed the result. K.S., Y.O., T.T. and J.C. performed experiments. T.F. wrote the manuscript.

## Additional information

**Competing interests:** The authors declare no competing interests.

