## [Peer Review File · Nature Communications]

Reviewers' Comments:

Reviewer #1:

Remarks to the Author:

This manuscript by Soltysik et al investigates the biogenesis and physiological function of nuclear Lipid Droplets in hepatocytes. Lipid droplets are ubiquitous cellular organelles, usually found in the cytoplasm. In some cell types, including hepatocytes, some Lipid Droplets are also found in the nucleus. Hepatocytes also have the capability to form Lipid Droplets in the lumen of the ER, which serve as precursors for the assembly of VLDL for secretion.

The current manuscript extends prior work by the Fujimoto laboratory to describe the pathway of nuclear LDs formation and to ascribe a physiological function in regulating PC synthesis to nuclear Lipid Droplets. Surprisingly the authors find that nuclear Lipid Droplets form by a mechanism similar to luminal Lipid Droplets, requiring for instance microsomal transfer protein (MTP), and an unconventional breaking through the inner nuclear envelope membrane into the nucleoplasm.

Overall, the paper is well done and the data are of high quality (although often alternative interpretation are possible, and controls are minimal). Publication of this manuscript will definitely drive the field by stimulating debate, and I thus support this.

A concern is that there are two and a half messages in the paper, and maybe as a consequence they appear a bit underdeveloped (one could see this to be two papers: one on biogenesis and one on the regulatory function of nuclear Lipid Droplets). In my view, the following points need to be addressed before publication:

1. While I think the "breaking through the inner nuclear membrane" is a really intriguing possibility, consistent with the data, I fail to be convinced that this happens. To make a strong claim as the authors do, they should provide additional data to support this. For the current data it remains possible that the EM images are explained by fixation artefacts and that the movies report on association of nuclear ER with already nuclear LDs. Is it a prediction from the authors model that there is a Ca^{++} burst at the site of INM rupture? Also could the authors use a lumenally expressed and a nucleoplasmic split fluorophore to visualize the rupture site? Alternatively, the authors could also really formulate their conclusions more carefully, saving more of the model for discussion and separate the data more clearly from the interpretations.

2. In my view, the part on the regulation of PC synthesis by nuclear Lipid Droplets is underdeveloped to some degree. For instance, the analysis of Tip47 and CCTa overlap on Lipid Droplets should be reconsidered as a binary scoring is very sensitive to the detection limit (this will not affect the authors conclusions, but is important). More importantly, for a number of assays on this point, controls are sparse. For example, it would be good to see controls for the assay with PC click labeling, and to perform more traditional PC synthesis assays (I understand this might be limited by transfection efficiency). For expression of different Tip47 constructs, quantitation of expression level is crucial to allow for comparing conditions (could be done by microscopy on the cellular level); also in these conditions, it would be good to show that the increase of PC synthesis is due to CCTa activation. For the data on MTP inhibition, it should be noted that this likely diminishes secretion of PC (as VLDL-like particles; a variable that would be best to address under all the conditions) which then indirectly could affect the need for more PC synthesis.

3. The link between Tip47 and ER stress is interesting but the link to nuclear LDs is really very, very weak. In my view, this could be left out and further explored in a different manuscript. Otherwise this would need a lot of controls and new experiments to make the link specifically to nuclear LDs; currently the data say that Tip47 has a weak effect on UPR activation (also these experiments lack some controls, such as TM alone, another UPR stimulator, secretion controls for ApoB (other proteins in the cell and such that are secreted,.....))

Reviewer #2:

Remarks to the Author:

Strengths: Work appears to be technically sound and sheds new light on the origin of nuclear LDs in hepatocytes. And furthermore makes a strong link between PC synthesis and the presence of perilipin 3 a perilipin family member for which a physiological function has been elusive. The paper thus sheds new light on lipid metabolism and regulation of lipid secretion in hepatocytes that should be of deep interest to the wider community. These results are highly novel and thus deserve publication after minor edits.

Weaknesses: The authors suggest that NR-luminal LDs often fused with nucleoplasmic LDs. They show EM images of partially merged LDs to prove their point. However, these images are NOT proof of the process of LD fusion. Instead they show just that, partial merger of LDs. Whether this is LD fusion or LD fission the authors have no way of distinguishing. A careful treatment of this data is warranted. In the discussion this issue comes back where the authors lay the link between LD fusion and the presence of supposedly more PE in the protein-phospholipid monolayer at the LD interface. This is perhaps plausible, but by no means a given. The authors might do well to cite work on the effect of PE (in vitro) on membrane fusion, but the process of LD fusion may be significantly different. A good paper for more information on the mechanism of LD fusion (in model systems) is the Langmuir paper by Ghimire et al from 2014 (Controlled particle collision leads to direct.....).

Point-by-point response to the referees' comments

Reviewer #1 (Remarks to the Author):

This manuscript by Soltysik et al investigates the biogenesis and physiological function of nuclear Lipid Droplets in hepatocytes. Lipid droplets are ubiquitous cellular organelles, usually found in the cytoplasm. In some cell types, including hepatocytes, some Lipid Droplets are also found in the nucleus. Hepatocytes also have the capability to form Lipid Droplets in the lumen of the ER, which serve as precursors for the assembly of VLDL for secretion.

The current manuscript extends prior work by the Fujimoto laboratory to describe the pathway of nuclear LDs formation and to ascribe a physiological function in regulating PC synthesis to nuclear Lipid Droplets. Surprisingly the authors find that nuclear Lipid Droplets form by a mechanism similar to luminal Lipid Droplets, requiring for instance microsomal transfer protein (MTP), and an unconventional breaking through the inner nuclear envelope membrane into the nucleoplasm. Overall, the paper is well done and the data are of high quality (although often alternative interpretation are possible, and controls are minimal). Publication of this manuscript will definitely drive the field by stimulating debate, and I thus support this.

Thank you for your kind words. We are truly happy to know that our paper was well received by you.

A concern is that there are two and a half messages in the paper, and maybe as a consequence they appear a bit underdeveloped (one could see this to be two papers: one on biogenesis and one on the regulatory function of nuclear Lipid Droplets). In my view, the following points need to be addressed before publication:

1. While I think the “breaking through the inner nuclear membrane” is a really intriguing possibility, consistent with the data, I fail to be convinced that this happens. To make a strong claim as the authors do, they should provide additional data to support this. For the current data it remains possible that the EM images are explained by fixation artefacts and that the movies report on association of nuclear ER with already nuclear LDs. Is it a prediction from the authors model is that there is a Ca⁺⁺ burst at the site of INM rupture? Also could the authors use a lumenally expressed and a nucleoplasmic split fluorophore to visualize the rupture site? Alternatively, the authors could also really formulate their conclusions more carefully, saving more of the model for discussion and separate the data more clearly from the interpretations.

We appreciate this thoughtful comment. The rupture of the inner nuclear membrane is definitely a new and unexpected finding, and we agree that it must be addressed as clearly as possible. As the reviewer pointed out, it is very important to exclude the possibility that the seemingly NR-luminal LDs in live imaging (Supplementary Video 1) were nucleoplasmic LDs and to show that the inner nuclear membrane rupture actually occurred using an alternative means. For this purpose, we carried out live imaging of perilipin-3-EGFP, a soluble protein in the nucleoplasm, and found that perilipin-3-negative luminal LDs became perilipin-3-positive immediately after the disruption of the NR-ring surrounding the LD (Supplementary Video 3 in the revised manuscript). This result demonstrated that the luminal LD becomes accessible to perilipin-3 only after the rupture of the NR membrane, further supporting our conclusion. Nevertheless, we cannot rule out the possibility that some nucleoplasmic LDs form by other mechanisms, as suggested by the reviewer. We thus revised the manuscript to leave this possibility open.

2. In my view, the part on the regulation of PC synthesis by nuclear Lipid Droplets is underdeveloped to some degree. For instance, the analysis of Tip47 and CCT α overlap on Lipid Droplets should be reconsidered as a binary scoring is very sensitive to the detection limit (this will not affect the authors conclusions, but is important).

In accordance with the reviewer's advice, we quantified the labeling intensities of perilipin-3 and CCT α in individual nuclear LDs and analyzed their correlation. As shown in the scatter plot (Supplementary Figure 6c in the revised manuscript), a majority of nuclear LDs are labeled for either of the proteins predominantly. Based on this plot, we classified nuclear LDs in three groups: perilipin-3-dominant (the relative intensity of perilipin-3/CCT α > 3), CCT α -dominant (the relative intensity of perilipin-3/CCT α < 1/3), and the other. The Venn diagram was revised to show the proportion of these three nuclear LD groups and is presented in Figure 6a. The result is basically the same as shown in the original manuscript, which used the binary classification, but the revised version shows the result more objectively. We are grateful for this suggestion by the reviewer.

More importantly, for a number of assays on this point, controls are sparse. For example, it would be good to see controls for the assay with PC click labeling, and to perform more traditional PC synthesis assays (I understand this might be limited by transfection efficiency).

We appreciate this helpful comment. To show the specificity of the PC click labeling, the results of several control experiments were added as Supplementary Figure 7a. They are: i) cells not incubated with propargylcholine, ii) cells incubated with propargylcholine together with an excess choline, iii) cells incubated with propargylcholine but treated with the click reaction mixture without Cu^+ . All of these samples had only negligible fluorescence signal, clearly showing the specificity of the method. The result of the click labeling method shows good correlation with that of the traditional ^3H -choline incorporation assay, as shown in Supplementary Figure 7a. As noted by the reviewer, however, it was difficult to use the ^3H -choline incorporation assay to examine the effect of transient perilipin-3 expression due to the low transfection efficiency in Huh7 cells. Therefore, as explained in the answer to the next question, we added a further control experiment using knockdown of CCT α .

For expression of different Tip47 constructs, quantitation of expression level is crucial to allow for comparing conditions (could be done by microscopy on the cellular level); also in these conditions, it would be good to show that the increase of PC synthesis is due to CCT α activation.

The expression levels of wild-type perilipin-3-V5 and the NES (nuclear exclusion signal)-appended perilipin-3-V5 were compared by measuring the fluorescence intensity given by the anti-V5 antibody labeling and also by Western blotting. The result showed that the perilipin-3-NES construct was expressed in a significantly higher level than the wild-type perilipin-3 (Supplementary Figure 7b). Nevertheless, only the wild-type perilipin-3 suppressed the PC synthesis significantly in comparison to non-transfected cells, indicating that perilipin-3 in the nucleus is crucial for the observed effect. Furthermore, when CCT α was knocked down by RNAi, the expression of either wild-type perilipin-3 or perilipin-3-NES did not cause significant change in PC synthesis, verifying that the effect of the perilipin-3 expression occurred through activation of CCT α . This result was added to the revised manuscript as Supplementary Figure 7c.

For the data on MTP inhibition, it should be noted that this likely diminishes secretion of PC (as VLDL-like particles; a variable that would be best to address under all the conditions) which then indirectly could affect the need for more PC synthesis.

We agree with the reviewer that MTP inhibition suppresses VLDL secretion, thereby decreasing the need for PC synthesis. A direct effect of MTP inhibition is a decrease in both ApoB-containing particles and ApoB-free luminal LDs, which is caused by

suppression of the lipid transfer in the ER. Our results showed that the ApoB-free luminal LD is the precursor of nucleoplasmic LDs, where CCT α is recruited to and activated, so that the decrease in ApoB-free luminal LDs by MTP inhibition also leads to a decrease in PC synthesis. This is thought to be a mechanism to regulate the rate of PC synthesis in accordance with that of VLDL secretion, and clearly explains the effect of MTP inhibition. These points were discussed in the revised manuscript.

3. The link between Tip47 and ER stress is interesting but the link to nuclear LDs is really very, very weak. In my view, this could be left out and further explored in a different manuscript. Otherwise this would need a lot of controls and new experiments to make the link specifically to nuclear LDs; currently the data say that Tip47 has a weak effect on UPR activation (also these experiments lack some controls, such as TM alone, another UPR stimulator, secretion controls for ApoB (other proteins in the cell and such that are secreted,.....))

We appreciate this helpful comment. The part addressing the effect of perilipin-3 knockdown on the ER stress/unfolded protein response was deleted from the revised manuscript.

Reviewer #2 (Remarks to the Author):

Strengths: Work appears to be technically sound and sheds new light on the origin of nuclear LDs in hepatocytes. And furthermore makes a strong link between PC synthesis and the presence of perilipin 3 a perilipin family member for which a physiological function has been elusive. The paper thus sheds new light on lipid metabolism and regulation of lipid secretion in hepatocytes that should be of deep interest to the wider community. These results are highly novel and thus deserve publication after minor edits.

Thank you for your kind words. We were most pleased to read that you believe our manuscript is worthy of publication following some modifications.

Weaknesses: The authors suggest that NR-luminal LDs often fused with nucleoplasmic LDs. They show EM images of partially merged LDs to proof their point. However, these images are NOT proof of the process of LD fusion. Instead they show just that, partial merger of LDs. Whether this is LD fusion or LD fission the authors have no way of distinguishing. A careful treatment of this data is warranted. In the discussion this issue comes back where the authors lay the link between LD fusion and the presence of supposedly more PE in the protein-phospholipid monolayer at the LD interface. This is perhaps plausible, but by no means a given. The authors might do well to cite work on the effect of PE (in vitro) on membrane fusion, but the process of LD fusion may be significantly different. A good paper for more information on the mechanism of LD fusion (in model systems) is the Langmuir paper by Ghimire et al from 2014 (Controlled particle collision leads to direct.....).

Thank you for this useful comment. The EM images show that the internal contents of luminal LDs and nucleoplasmic LDs are continuous with each other, but as pointed out by the reviewer, these images alone are not sufficient to conclude that the LDs are in the fusion process. We thus revised the related text in the Results section. We also deleted the sentence in the Discussion section that speculated on the relevance of rich PE for LD fusion. We believe that the behavior of nuclear LDs is an important issue and that it should be studied in future research.

Reviewers' Comments:

Reviewer #1:

Remarks to the Author:

This manuscript is thoughtful and though provoking. The authors have carefully addressed the concerns raised during the revision. I now fully support publication of this work in Nature Communications.

Reviewer #2:

Remarks to the Author:

In this revision the authors have now addressed all of my concerns. This is an interesting manuscript that sheds some light on the potential function of perilipin 3, an enigmatic perilipin family member.

Dr. ing. Edgar E. Kooijman

REVIEWERS' COMMENTS:

Reviewer #1 (Remarks to the Author):

This manuscript is thoughtful and though provoking. The authors have carefully addressed the concerns raised during the revision. I now fully support publication of this work in Nature Communications.

Reviewer #2 (Remarks to the Author):

In this revision the authors have now addressed all of my concerns. This is an interesting manuscript that sheds some light on the potential function of perilipin 3, an enigmatic perilipin family member.

Dr. ing. Edgar E. Kooijman

We thank the reviewers for constructive comments, which helped to improve the paper considerably.